# Can Hematological Parameters Play a Role in the Differential Diagnosis of Adrenal Tumors?



Mehmet Gürkan Arıkan [1,*] , Göktan Altuğ Öz [1], Nur Gülce İşkan [2] , Necdet Süt [3], İlkan Yüksel [4] and Ersan Arda [1]

1. Department of Urology, Trakya University School of Medicine, 22030 Edirne, Turkey; goktanoz@yandex.com (G.A.Ö.); ersanarda@gmail.com (E.A.)
2. School of Medicine, Trakya University, 22030 Edirne, Turkey; nurgulceiskan@gmail.com
3. Department of Biostatistics and Medical Informatics, Trakya University School of Medicine, 22030 Edirne, Turkey; nsut@trakya.edu.tr
4. Department of Urology, Private Kesan Hospital, 22800 Edirne, Turkey; ilkanyuksel@hotmail.com
* Correspondence: mgarikan26@gmail.com or mgurkanarikan@trakya.edu.tr; Tel.: +9-050-7929-7204

**Abstract:** There have been few studies reported with conflicting results in the use of neutrophil-lymphocyte ratio (NLR), platelet-lymphocyte ratio (PLR), redcell-distribution-width (RDW), etc. for predicting prognosis and differential diagnosis of adrenal tumors. The aim of this study is to investigate the role of inflammatory markers through a complete blood count, which is an easy access low-cost method, for the differential diagnosis of adrenocortical adenoma (ACA), adrenocortical carcinoma (ACC), and pheochromocytoma. The data of patients who underwent adrenalectomy between the years 2010–2020 were retrospectively analyzed. Systemic hematologic inflammatory markers based on a complete blood count such as neutrophil ratio (NR), lymphocyte ratio (LR), NLR, PLR, RDW, mean platelet volume (MPV), and maximum tumor diameter (MTD) were compared between the groups. A statistically significant difference was found between the three groups in terms of PLR, RDW, and MTD. With post-hoc tests, a statistically significant difference was found in PLR and MTD between the ACA and ACC groups. A statistically significant difference was found between the ACA and pheochromocytoma groups in PLR and RDW values. In conclusion, it could be possible to plan a more accurate medical and surgical approach using PLR and RDW, which are easily calculated through an easy access low-cost method such as a complete blood count, together with MTD in the differential diagnosis of ACC, ACA, and pheochromocytoma.

**Keywords:** adrenal tumors; hematological markers; inflammation; neutrophil-lymphocyte ratio; platelet-lymphocyte ratio; red cell distribution width

## 1. Introduction

Adrenal tumors are seen 3–10% in the general population and the majority are made up of benign adrenocortical adenomas (ACA) [1]. Although most ACAs are non-functional, approximately 15% have been reported to produce hormone secretion [2]. On the other hand, adrenocortical carcinomas (ACC) are very rarely seen in 1–2 patients per million in a year, and approximately 40–60% show hormonal hypersecretion (Cushing's syndrome, hyperaldosteronism, etc.) [3]. In addition, pheochromocytomas are other rare adrenal tumors originating from the adrenal medulla that secrete catecholamines [4]. All three tumors are generally asymptomatic, although some may exhibit clinical findings associated with endocrinopathies due to excessive hormone secretion, abdominal pain may be a symptom depending on the size of the mass, as well [1]. Diagnosis is based on adrenal imaging as well as clinical and laboratory (hormonal) evaluation [5]. Differential diagnosis of functional ACA and ACC may be difficult due to hormonal hypersecretion [6,7]. Even though new imaging techniques have been progressively developing in recent years, they have been reported to be insufficient [5]. In addition to fine-needle aspiration biopsy being

an invasive procedure, it has been seen to have a low effect in differentiating primary benign and malignant tumors of the adrenal gland [8]. All of these factors highlight the importance of using new methods for differential diagnosis.

Interest in the relationship between inflammation and cancer has been increasing throughout recent years. Cancer-inflammation-related processes in tumors are associated with the production of cytokines (IL-6 and TNF-$\alpha$), growth factors, and granulocyte colony-stimulating factors in oncology patients [3]. The systemic inflammatory responses are associated with thyroid, pancreas, and many endocrine organ malignancies, as well as in urogenital malignancies such as prostate and testis [9–12]. Furthermore, the association of inflammatory response with hypertension, diabetes mellitus, and many systemic diseases has also been demonstrated [13]. Cushing disease and hyperaldosteronism, which cause excessive secretion of adrenal hormones, lead to an increase in inflammation by affecting the immune system [14,15]. However, immunohistochemically, it has been reported that inflammation around the tumors' microenvironment is seen at a lower level in ACC rather than ACA and pheochromocytoma [16].

Hematological parameters such as neutrophil-lymphocyte ratio (NLR), platelet-lymphocyte ratio (PLR), red cell distribution width (RDW), which are used as indicators of acute and chronic inflammation, are effective in the diagnosis and prediction of inflammatory disease and some malignancies [3,11,17,18]. There have been few studies reported with conflicting results in the use of NLR, PLR, RDW, etc. for predicting prognosis and differential diagnosis of adrenal tumors. Hence, the aim of this study is to investigate the role of inflammatory markers through a complete blood count, which is an easy access low-cost method, for the differential diagnosis of ACA, ACC, and pheochromocytoma.

## 2. Materials and Methods

### 2.1. Data Extraction

The data of patients who underwent adrenalectomy between the years of 2010–2020 were retrospectively analyzed from the hospital electronic system in the Urology Clinic. Analyzed data contain the patients' basic demographic characteristics (age, gender, etc.) as well as blood test results that include routine biochemical and radiological parameters. Informed consent forms were obtained from the patients and the study was approved by the university's scientific research ethics committee.

### 2.2. Patient Selection

The including criteria consist of patients whose pathological evaluation of adrenalectomy material resulted in ACC, ACA, and pheochromocytoma. Patients who underwent adrenalectomy during radical nephrectomy, who underwent adrenalectomy due to adrenal metastasis, who had an active infection, an inflammatory (ankylosing spondylitis, FMF), and a hematological disease affecting their hemogram parameters were excluded from the study. Systemic hematologic inflammatory markers based on a complete blood count such as neutrophil ratio (NR), lymphocyte ratio (LR), NLR, PLR, RDW, MPV were compared between the groups. NLR was found by dividing the neutrophil count to the lymphocyte count, whereas PLR was found by dividing the platelet count to the lymphocyte count. In addition, the maximum tumor diameter (MTD) was measured by computed tomography and the groups were compared.

### 2.3. Statistical Analysis

Data analysis was performed using IBM SPSS v 23.0.0 (IBM Corp., Armonk, NY, USA). Numbers, percentages, arithmetic mean, and median values were evaluated as descriptive statistics. A *p*-value < 0.05 was set for statistical significance. The data were divided into three groups as ACC, ACA, and pheochromocytoma. The chi-square test was used for the comparison of categorical variables (gender). The variables were tested for normal distribution with the Shapiro-Wilk test. The variables within a normal distribution were compared using the ANOVA test, and non-normally distributed variables were compared

with the Kruskal-Wallis test. A post-hoc test using Dunn's test with Bonferroni correction was used to determine the statistically significant difference between the groups and the significance value was adjusted the as $p < 0.05$.

## 3. Results

After the inclusion and exclusion criteria were taken into evaluation, 52 adrenalectomy patients remained. Patients were divided into three groups, 32 out of 52 patients were in ACA, 12 were in ACC, and eight were in the pheochromocytoma group. Age, MTD, NR, LR, NLR, PLR, RDW, and MPV values were compared. According to the distribution of the groups, the mean ages of the patients were $53.34 \pm 11.28$ years in the ACA group, $55.83 \pm 16.05$ years in the ACC group, $54.5 \pm 12.24$ years in the pheochromocytoma group. Descriptive statistics and their distribution by groups are summarized in Table 1.

**Table 1.** Statistics of patients' demographic data, hematological parameters, and maximum tumor diameter.

| Descriptive Statistics | ACA Mean $\pm$ Standard Deviation | ACC Mean $\pm$ Standard Deviation | Pheochromocytoma Mean $\pm$ Standard Deviation | *p*-Value | Normal Values |
|---|---|---|---|---|---|
| Age (years) | $53.34 \pm 11.28$ | $55.83 \pm 16.05$ | $54.5 \pm 12.24$ | 0.841 | |
| Gender (N (%)) | | | | | |
| -Male | 17 (53.1%) | 5 (41.7%) | 2 (25%) | | |
| -Female | 15 (46.9%) | 7 (58.3%) | 6 (75%) | 0.236 | |
| -Total | 32 (100%) | 12 (100%) | 8 (100%) | | |
| NLR (%) | 2.1 (1.28) * | 2.45 (6.77) * | 3.7 $\pm$ 2.25 | 0.105 | |
| PLR (%) | 106.75 (41.41) * | 149.4 (161.44) * | 164.99 (205.93) * | 0.001 | |
| NR (%) | 61.13 $\pm$ 8.24 | 64.11 $\pm$ 19.66 | 67.3 $\pm$ 9.46 | 0.603 | 34–67.9 |
| LR (%) | 28.71 $\pm$ 7.25 | 25.49 $\pm$ 15.45 | 21.54 $\pm$ 7.45 | 0.311 | 21.8–53.1 |
| MPV ($10^3$/uL) | 8.73 $\pm$ 1.27 | 8.65 $\pm$ 1.38 | 8.41 $\pm$ 1.06 | 0.836 | 9.4–12.4 |
| RDW ($10^3$/uL) | 14.1 (1.85) * | 15.15 (3.4) * | 16.5 (3.43) * | 0.031 | 11.6–14.4 |
| MTD (cm) | 3.85 (3.78) * | 7.8 (8.30) * | 4.5 (2.98) * | 0.031 | |

ACA: Adrenocortical adenoma; ACC: Adrenocortical carcinoma; NR: Neutrophil ratio; LR: Lymphocyte ratio; NLR: Neutrophil-lymphocyte ratio; PLR: Platelet-lymphocyte ratio; MPV: Mean platelet volume; RDW: Red cell distribution width; MTD: Maximum tumor diameter. * on-normally distributed variables were given as median (interquartile range).

A statistically significant difference was found between the three groups in terms of PLR, RDW, and MTD ($p = 0.001$, $p = 0.031$, $p = 0.031$, respectively) (Table 1). With post-hoc tests, a statistically significant difference was found in PLR and MTD between the ACA and ACC groups ($p = 0.015$, $p = 0.035$, respectively). A statistically significant difference was found between the ACA and pheochromocytoma groups in PLR and RDW values ($p = 0.005$, $p = 0.047$, respectively) (Table 2).

**Table 2.** Post-hoc test results of patients.

| | ACA vs. ACC | ACA vs. Pheochromacytoma | ACC vs. Pheochromacytoma |
|---|---|---|---|
| PLR | $p = 0.015$ | $p = 0.005$ | $p = 1$ |
| RDW | $p = 0.335$ | $p = 0.047$ | $p = 1$ |
| MTD | $p = 0.035$ | $p = 0.566$ | $p = 1$ |

PLR: Platelet-lymphocyte ratio; RDW: Red cell distribution width; MTD: Maximum tumor diameter.

## 4. Discussion

Our study showed that PLR and MTD values were significantly higher in ACC than ACA, whereas RDW and PLR values were significantly higher in pheochromocytomas than ACA. These findings demonstrate that MTD, PLR, and RDW values can be used in the differentiation of ACA-ACC-pheochromocytoma. Thus far, two studies have investigated hematological markers in the differential diagnosis of ACA and ACC [19,20].

In the most recent study by Şişman et al. [20] which was conducted with the participation of 39 patients, PLR was found statistically significant between adenoma and

carcinoma groups as well as our study. Recent studies show that the platelets' concentration in the tumor microenvironment could play a significant role in stimulating tumor development [21]. Thrombocytosis seen in many patients with solid tumors is associated with tumor infiltration and metastasis [21]. This information supports the statistically significant difference of PLR value between ACA-ACC groups in our study.

Şişman et al. [20] found a statistically significant difference in NLR values between ACA and ACC groups. Another study by Mochizuki et al. [19], consisting of 59 patients performed in 2017, also found that NLR was effective in distinguishing ACC from non-malignant adrenal tumors. However, in our study, NLR was found to be statistically insignificant in the differential diagnosis. This contradiction in NLR values between our study and the literature may be caused due to the sample size. Since adrenocortical tumors are rare, gathering a homogenous sample size with a greater number of patients is challenging. Therefore, in all of the studies, including ours, the population size may not be enough to reflect the real-world data. Further studies with a greater sample size and diverse patient populations are needed to investigate the role of NLR in adrenocortical tumors. It has been reported in the literature that some functional adrenocortical tumors increase the neutrophil count and decrease lymphocytes [14]. Unlike the two studies above, all of our patients in the ACA group have functional adenomas and although the NLR values are not as high as ACC, the increase in the ACA group, due to the functionality of the tumors, may play a role in the difference between our study and the literature.

In recent years, developments in cancer immunology show the crucial role of cells in the tumor microenvironment. Utilization of this information can have prominent effects on establishing patients' prognostic criteria. Guadagno et al. [16] demonstrated how lymphocytes surrounding ACA were higher compared to ACC, which was supported by the possible immunosuppressive role of malignant tumor cells. These results were also associated with a better prognosis [16]. Supporting these results, studies investigating prognostic markers of ACC show how PLR values are higher in patients with poor prognosis [3,16,22]. Due to the retrospective nature of our study, we could not analyze the prognosis of our patients. However, higher PLR values detected in the ACC group, in addition to imaging modalities and other laboratory tests, may help with the benign and malignant behavior of adrenal tumors.

In addition, it was shown in the study by Guadagno et al. [16] that the immune response in pheochromocytoma, a tumor originating from the adrenal medulla, is not different from other adrenal tumors. Our study differs from this study by the statistical difference in PLR and RDW values between ACA and pheochromocytoma groups. The immunohistochemical study by Guadagno et al. [16] provides information about the local immune reaction and found no difference between ACA and pheochromocytoma. A systemic response was at the forefront in our study, and the fact that catecholamines trigger the systemic immune response may significantly make the difference between these two studies. It has been shown that excessive catecholamine secretion in pheochromocytoma triggers systemic inflammation and increases acute phase reactants, as well as increases active platelet counts more than the lymphocyte count [4]. This may explain the higher detection of PLR levels in pheochromocytoma compared to ACA. On the other hand, many carcinomas and systemic inflammatory diseases have also been shown to increase RDW levels in parallel with inflammatory markers [18]. The increase in RDW in parallel with the increased acute phase reactants in excess catecholamine secretion supports the difference in the pheochromocytoma and ACA groups in our study. Although RDW is a complex marker that can be affected by factors such as age and diet, to our knowledge for the first time in the literature, we argue that RDW may be a new supportive marker in addition to the tests used to differentiate ACA from pheochromocytoma [18].

In the studies that investigate the malignancy and prognostic values of the tumor size in adrenal incidentalomas, it has been seen that when the tumor size increases (especially masses of 4–6 cm and above), the transformation into malignancy also increases, and the prognosis worsens [5,19]. In our study, the median MTD in the ACC group was found as

7.8 cm, almost twice as the ACA group, which is consistent with the literature [5,19]. Therefore, our findings demonstrate that the combination of hematological parameters and tumor size could be effective in the diagnosis and follow-up of patients with adrenal tumors.

The main limitation of our study was its retrospective nature and limited patient population. Another limitation was the inadequate homogenous distribution between the groups due to the rarity of ACC in general. Further multicentered randomized control studies are needed to investigate the use of new markers in the differential diagnosis of ACC that could later on be included in future guidelines.

## 5. Conclusions

In conclusion, PLR and RDW may be two new markers which can be easily calculated based on an easy-access low-cost method, such as a complete blood count, that can help in the differentiation of ACC, ACA, and pheochromocytoma with a comprehensive and detailed approach.

**Author Contributions:** Conceptualization, E.A. and M.G.A.; methodology, N.G.İ. and M.G.A.; software, G.A.Ö.; validation, N.G.İ., M.G.A., and G.A.Ö.; formal analysis, N.S.; investigation, M.G.A.; resources, G.A.Ö. and N.G.İ.; data curation, N.G.İ.; writing—original draft preparation, M.G.A.; writing—review and editing, İ.Y., E.A., and N.S.; visualization, N.S. and İ.Y.; supervision, İ.Y. and E.A.; project administration, M.G.A. and G.A.İ.; funding acquisition, none. All authors have read and agreed to the published version of the manuscript.

**Funding:** This research received no external funding.

**Institutional Review Board Statement:** The study was conducted according to the guidelines of the Declaration of Helsinki, and approved by the Ethics Committee of Trakya University, School of Medicine (TUTF-BAEK 2021/204 and 15.03.2021).

**Informed Consent Statement:** Informed consent was obtained from all subjects involved in the study.

**Data Availability Statement:** The data presented in this study are available on request from the corresponding author. The data are not publicly available due to restrictions of privacy.

**Conflicts of Interest:** The authors declare no conflict of interest.

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
