# Peer review of "Can Hematological Parameters Play a Role in the Differential Diagnosis of Adrenal Tumors?"

_2673-4397, doi:10.3390/uro1020006_

Round 1

Reviewer 1 Report

It was a pleasure to review this paper, “Can Hematological Parameters Play a Role in the Differential
 Diagnosis of Adrenal Tumors?”. 
Adrenal tumors are rare and heterogeneous diseases, and scientific researches are welcome in this context. 

I have some concerns about this paper:

*Line 104: There is a wrong p-value. Please correct it. 
*Line 156: I did not understand why a retrospective study could not give information about prognosis. In ACC cohorts, even with one-year follow-up is possible to determine an aggressive tumoral behavior. Most cases of ACA are benign and rare cases present recurrence. 
It is essential to clarify the reason for not presenting a prognosis value for the hematological parameters. Are there missing data in the medical record?  
*Line 157: Although there is a difference in PLR values between ACC and ACA groups, knowing and being a researcher in the adrenal field, it could not be possible to use in medical assistance the hematological parameters in the differentiation of benign and malignant adrenal tumors. 
There is a comprehensive and detailed approach to determine if an adrenal tumor is a malignant or benign tumor (see ref: ENSAT- doi: 10.1530/EJE-18-0608). I suggest correcting this statement due to the complexity of the disease. I would affirm that PLR could be an additional item with a value of benignity, which corroborates in the context with other markers to determine the tumoral behavior.

Line 176 and 177:  I suggest rewriting this statement due to the same reasons as Line 157. 
Line 185: I suggest completing the conclusion, affirming that PLR and RDW could be two new benignity markers, which could help differentiate ACC from ACA within a comprehensive and detailed approach. 

Author Response

Dear Uro Editorial Office: 

Many thanks for your e-mail note on April 26, 2021, and we are most grateful to the editorial office and the reviewers for the helpful comments on the manuscript. We have taken all these comments into account and submit the revised version of our paper. Please see the attachment.

We focused the critique based on the reviewers’ excellent comments point by point and made highlight in this revised manuscript. 

We would like to express our gratitude for the help we received from the reviewer’s comments. We hope that the revisions of our work are satisfactory, and we hope that the revised version of our paper is now suitable for publication in the prestigious journal ‘’Uro’’ and we look forward to hearing from you at your earliest convenience. 
Thank you very much for reviewing this submission. 

Yours sincerely, 

Reviewer 2 Report

In materials and method paragraph should be indicated how NLR and PLR have been calculated.

It also should be specified which radiological investigations were evaluated for MTD.

Lines 93-94: p-value for differences between groups should be adjusted according to post hoc test comparison, pointing which test was adopted since there are three groups in this study. The adjusted p-value level should be indicated.

Lines 100-101: the mean ages indicated do not correspond to those in Table 1

Table 1: unit of measure for NR, LR, MPV and RDW should be indicated. It would be also useful to describe in table 1 the number of subjects for each group. Also, add a column for normal values would be helpful for the interpretation of results.

Line 127: the semicolon should be a comma

Lines 159-162: This part is not very clear: you should better explain why you found a significative difference between Pheocromocytoma and ACA while Guadagno et al. showed no differences between tumors of adrenal medulla and other adrenal tumors.

This paper contains some findings which could be very helpful in differential diagnosis of adrenal tumors, so it should be accepted after these minor revisions.

Author Response

(The authors gave the same response as above.)

Round 2

Reviewer 1 Report

Dear authors, 
Congratulations!

This manuscript is ready for being published. 
Success to your team!